# Soil is a major contributor to global greenhouse gas emissions and climate change

Peter M. Kopittke[*,1], Ram C. Dalal[1], Brigid A. McKenna[1], Pete Smith[2], Peng Wang[3], Zhe (Han) Weng[1], Frederik J.T. van der Bom[1], and Neal W. Menzies[1]

[1]The University of Queensland, School of Agriculture and Food Sustainability, St Lucia, Queensland, 4072, Australia.

[2]Institute of Biological and Environmental Sciences, University of Aberdeen, Aberdeen, AB24 3UU, Scotland, United Kingdom.

[3]Nanjing Agricultural University, College of Resources and Environmental Sciences, Nanjing 210095, China.

*Correspondence to*: Peter M. Kopittke (p.kopittke@uq.edu.au)

**Abstract.** It is unequivocal that human activities have increased emissions of greenhouse gases, that this is causing warming, and that these changes will be irreversible for centuries to millennia. Whilst previous studies have broadly examined the contribution of agriculture or land-use change to anthropogenic greenhouse gas emissions, the contribution of soil itself remains unclear, with quantifying the contribution of soil in this regard being critical for developing and implementing appropriate management practices. In the present study, we used previously published datasets for carbon dioxide, nitrous oxide, and methane to determine soil-based emissions of greenhouse gases and their contribution to anthropogenic greenhouse gas emissions. We show that our near-complete reliance on soil to produce the rapidly increasing quantities of food being demanded by humans has caused soil to release profound amounts of greenhouse gases that are threatening the future climate. Indeed, net anthropogenic emissions from soil alone account for 15% of the entire global increase in climate warming (radiative forcing) caused by well-mixed greenhouse gases, with carbon dioxide being the most important gas emitted from soil (74% of total soil-derived warming) followed by nitrous oxide (17%) and methane (9%). There is an urgent need to prevent further land-use change (including for biofuel production) to limit the release of carbon dioxide that results from loss of soil organic carbon, to develop strategies to increase nitrogen fertilizer efficiency to reduce nitrous oxide emissions, to decrease methane from rice paddies, and to ensure that the widespread thawing of permafrost is avoided. Innovative approaches are urgently required for reducing greenhouse gas emissions from soil if we are to limit global warming to 1.5 or 2.0 °C.

## 1 Introduction

Soil is multifunctional and provides a diverse range of services. One important role of soil is in producing 98.8% of the calories consumed by humans – 12.2% (1,556 million ha) of global ice-free land is used for cropping and 24.8% for grazing (FAO, 2021). Given that the vast majority of human food comes from soil, profound changes in land-use over the history of agrarian society has greatly increased stresses on soil (Kopittke et al., 2019). The ongoing increasing demand for food is due to both a rapidly increasing population, from 2.5 billion in 1950 to 7.8 billion in 2020 (projected to be 9.8 billion by 2050), and to increasing rates of consumption per capita. There are also other demands on soil, including land-use for bioenergy production, with land devoted for biofuel production increasing from 7 million ha in 2000 to 32 million ha in 2010 (Langeveld et al., 2013).

The reliance of humans on soil is causing the substantial release of anthropogenic greenhouse gases, especially carbon dioxide ($CO_2$), nitrous oxide ($N_2O$), and methane ($CH_4$), contributing markedly to climate change. Climate change is the greatest challenge facing human society, and it is "unequivocal that human influence has warmed the atmosphere, ocean and land" and that many of the resulting changes will be "irreversible for centuries to millennia" (IPCC, 2021). Thus, our need to rapidly

increase food production from soil whilst simultaneously decreasing the greenhouse gas emissions associated with this production represents a 'wicked problem' (Rittel and Webber, 1973). If we continue to solely focus on the role of soil to provide humans with food without recognizing, and acting upon, its profound contributions to greenhouse gas emissions and climate change, we will threaten the hospitability of our planet for millennia and fail to recognize intergenerational equality.

Soil acts as both a source and a sink for natural and anthropogenic greenhouse gases. For example, for C, the net global input of C to soil from vegetation is ca. 61 Pg C /y, with a similar amount lost from soil to the atmosphere as $CO_2$ (Lehmann and Kleber, 2015). However, anthropogenic use of soil and a changing climate have altered this natural balance. For example, it is known that boreal and temperate forests of the northern hemisphere are making an increased contribution to the terrestrial (vegetation plus soil) C sink (Canadell et al., 2021), with these systems having increased biomass production due to $CO_2$ fertilization and lengthening growing seasons. Nevertheless, it is also known that soil globally contains ca. 116 Pg of C less now than prior to land-use change (Sanderman et al., 2017), indicating that despite these localized regions of increased C sequestration in soil, there has been an overall net global decrease in global C stocks and hence a net release of $CO_2$ to the atmosphere. In a similar manner, soil is both a source and a sink for $CH_4$ – soil acts as a sink for ca. 30 Tg $CH_4$ /y, with this representing ca. 4% of total $CH_4$ emissions in 2017 (Saunois et al., 2020). However, soil is also both a natural and anthropogenic source of atmospheric $CH_4$ – use of soil for rice cultivation, for example, also accounts for 30 Tg $CH_4$ /y (Saunois et al., 2020). Thus, despite the ongoing critical role of soil as a sink for greenhouse gases, it is also imperative to quantify how the anthropogenic use of soil has also increased atmospheric emissions of greenhouse gases from soil. This is because the net anthropogenic increase in emissions from soil, together with emissions of greenhouse gases from other sources such as burning of fossil fuels, also contribute to global warming and climate change.

The aim of the present study was to quantify the contribution of soil to global anthropogenic greenhouse gas emissions and global climate change. To the best of our knowledge, no studies have reported this information previously for soil. Although Oertel et al. (2016) examined the rate at which greenhouse gases are evolved from soil, these authors did not consider the overall net contribution of soil to climate change, whilst other studies have examined the contribution of agriculture more broadly (Robertson et al., 2000; Jia et al., 2019; Amundson, 2022). In a similar manner, previous studies have examined the contributions of agriculture, forestry and other land use (AOLU) to greenhouse gas emissions. However, in order to improve management practices and to inform better decision-making processes, it is imperative that we quantify the precise sources of greenhouse gases and understand the factors causing their emissions. In this regard, we differentiate here between 'soil' and 'land', with soil being a core nested component of 'land' while 'land' has a broader context consisting of soil, rocks, rivers, and vegetation (Koch et al., 2013) – it is necessary to distinguish between 'soil' and 'land' to understand, value, and manage soil as a discrete component of the broader landscape. Our work also complements the increasing number of studies that examine the potential of soil as a nature-based solution to $CO_2$ removal and climate change mitigation (Smith, 2012; Paustian et al., 2016; Minasny et al., 2017; Lal et al., 2021; van Vuuren et al., 2018; Crow and Sierra, 2022). We need to first accurately quantify the substantial quantities of anthropogenic greenhouse gas emissions from soil and their contribution to climate change before we can properly estimate the potential of soil to mitigate greenhouse gas emissions. We show that soil is a major contributor to global greenhouse gas emissions and that there is a need to urgently improve management of soil if we are to simultaneously increase food production whilst also limiting global climate change.

## 2 Materials and Methods

This study examined anthropogenic, soil-based emissions of greenhouse gases and their contribution to climate change. All underlying data used here were derived from previous studies (see later). For all three greenhouse gases, we examined the

contribution of soil to emissions using two broad approaches. The first was to examine how the current annual net anthropogenic flux from soil compares to the total anthropogenic flux from all sources to determine the current soil-derived contributions to current greenhouse gas emissions. The second approach was to calculate the contribution of soil-based emissions to the current increase in effective radiative forcing due to anthropogenic greenhouse gases, with the increase in effective radiative forcing not only due to current fluxes, but also due to historical emissions. Currently, the total increase in effective radiative forcing due to increased concentrations of well-mixed greenhouse gases ($CO_2$, $CH_4$, $N_2O$, and halocarbons) is +3.32 $W/m^2$, of which +2.16 $W/m^2$ is due to $CO_2$, +0.21 $W/m^2$ is due to $N_2O$, and +0.54 $W/m^2$ is due to $CH_4$ (Forster et al., 2021). [The overall net increase in effective radiative forcing when taking into account all climate forcers, including those which decrease effective radiative forcing such as aerosol cloud interactions, is +2.84 $W/m^2$] (Forster et al., 2021). For each of these three greenhouse gases, we calculate the total anthropogenic contribution of soil to the current increase in radiative forcing by using historical data. For this, we determine the proportion of total anthropogenic emissions that have been derived from soil over time whilst also taking into account the atmospheric life of the gas, with this calculating the proportion of the current increase in anthropogenic radiative forcing that is due to soil. For each of the three gases, the length of time over which anthropogenic emissions from soil were determined, as well as the data-intensity over that period, depended upon the data sources that were available (see below).

## 2.1 Carbon dioxide

For $CO_2$, to determine the anthropogenic soil-derived contributions to greenhouse gas emissions, we used the data of Sanderman et al. (2017) who modelled spatial changes in SOC stocks over time due to agriculture. By comparing changes in total global SOC stocks, as opposed to changes in net inputs or outputs from soil, this disentangles multiple confounding factors – if the global SOC stock is a given quantity lower (or higher) than the corresponding value prior to land-use change, it is unambiguous that this net mass of C must have been lost to (or sequestered from) the global atmosphere due to anthropogenic use of soil despite any potential increase in SOC sequestration rates in soils of particular areas where they are acting as a net sink. Sanderman et al. (2017) used a machine learning-based data-driven statistical model based upon soil profile observations, with this coupled with the History Database of the Global Environment (HYDE) (Sanderman et al., 2017).

Using the study of Sanderman et al. (2017), we used the values reported for cumulative loss of SOC over time [Pg C, Fig. 2 of Sanderman et al. (2017)] to calculate the rate of net decrease in SOC stocks (Pg C/y) and associated net emission of $CO_2$ (i.e. the first of the two approaches articulated above): the most recent data point of Sanderman et al. (2017) was used to determine the current annual net anthropogenic flux from soil whilst the entire data set [Fig. 2 of Sanderman et al. (2017)] was used to determine the total (historical) total anthropogenic contribution of soil. In addition, this historical assessment of the total anthropogenic contribution of soil to the current increase in radiative forcing (i.e. our second approach articulated above) requires consideration of the atmospheric life of the gas. However, given that $CO_2$ is chemically inert in the atmosphere, there is no single value for the atmospheric life of $CO_2$, but part of the $CO_2$ emitted by humans remains in the atmosphere for millennia (Forster et al., 2021). Rather, we simply determine the proportion of the cumulative anthropogenic emissions of $CO_2$ from all sources that was due to anthropogenic emissions from soil (being from land-use change and loss of SOC, as discussed later). In other words, to calculate the contribution of soil-based emissions to the current increase in effective radiative forcing due to anthropogenic greenhouse gases, we simply calculated the proportion of total historical anthropogenic $CO_2$ emissions that have been derived from soil by determining total cumulative net anthropogenic emissions from soil (Sanderman et al., 2017) with total cumulative anthropogenic emissions (Friedlingstein et al., 2023). Because we do not use a value for atmospheric life for $CO_2$, we therefore assume that historical anthropogenic emissions of $CO_2$ from soil contributes equally to increases in radiative forcing compared to the more recent emissions of $CO_2$ from fossil sources. In this regard, it must be

noted that although 57% of the $CO_2$ emitted into the atmosphere is absorbed by the ocean sink and the terrestrial sink (being 26% for the ocean sink and 31% for the terrestrial sink) (Friedlingstein et al., 2023), we have assumed that the proportion of $CO_2$ absorbed by sinks is constant for both the soil-based source and fossil sources despite the soil-based source occurring over a longer period of time, with this likely causing an over-estimate of the contribution of $CO_2$ emissions from soil to the current increase in radiative forcing. Indeed, as discussed later, emissions from soil have increased rapidly during in the last ca. 100-200 y, whilst in contrast, emissions from fossil sources have occurred primarily during the last ca. 60 to 70 y. Regardless, even for C that is absorbed by the ocean sink, although it does not remain in the atmosphere where it contributes to climate change, it causes ocean acidification which (like climate change) is also considered to be a critical Earth-system process (Steffen et al., 2015).

## 2.2 Nitrous oxide

For $N_2O$, we took a slightly different approach than that used for $CO_2$ where we examined the changes in global SOC stocks. Rather, for $N_2O$, we determined the proportion of total anthropogenic emissions that were due to anthropogenic emissions from soil. This provides data on the proportion of anthropogenic $N_2O$ being released to the atmosphere that is due to human-use of soil. For this, we used the data available from Tian et al. (2019) and Tian et al. (2020), with these authors using process-based models that considers C, N, and water cycling to simulate soil $N_2O$ emissions. For $N_2O$, we took into account the atmospheric life of $N_2O$ [the time to decrease to a concentration of 1/e, being 109 y, see Forster et al. (2021)]. Although the atmospheric life of $N_2O$ is 109 y, we are only able to calculate the proportion of total anthropogenic $N_2O$ emissions that have resulted from soil from 1860s onwards as we are unaware of data examining emissions from soil prior to this point. Nevertheless, $N_2O$ emissions were low prior to 1860s (Syakila and Kroeze, 2011), with the majority of $N_2O$ emissions being associated with the application of reactive N fertilizers, the usage of which increased profoundly from the 1950s and 1960s onwards (Erisman et al., 2008).

## 2.3 Methane

For $CH_4$, we used a similar approach as for $N_2O$ – we examined the proportion of the anthropogenic $CH_4$ emissions that have resulted from anthropogenic use of soil as rice paddies. This does not neglect the simultaneous role of soil as a sink for $CH_4$, but it determines the magnitude of the increase in atmospheric $CH_4$ due to human-use of soil. Given that the atmospheric life of $CH_4$ is 11.8 y (Forster et al., 2021), we only examined $CH_4$ emissions from 1980 onwards. We are unaware of data examining historical $CH_4$ emissions from rice paddies. Therefore, given that current emissions of $CH_4$ are 30 Tg per year [see Saunois et al. (2020)] from the 162 million ha of rice paddies (FAO, 2021), to estimate historical emissions from rice paddies, we assumed that the rate of release per hectare was constant and simply adjusted emissions based upon the area of rice paddies (FAO, 2021). These values for $CH_4$ from rice paddy soil were compared to corresponding values for total anthropogenic $CH_4$ emission since 1980 as reported by Saunois et al. (2020) and He et al. (2020).

## 3 Results

### 3.1 Carbon dioxide

Atmospheric concentrations of $CO_2$ have increased from 278 ppm in 1750 to 419 ppm in 2023 and with concentrations having increased from 391 ppm in 2011 to 419 ppm in 2023 alone (Gulev et al., 2021; Friedlingstein et al., 2023). Given that the total increase in radiative forcing due to anthropogenic increases in well-mixed greenhouse gas concentrations is +3.32 W/m$^2$, and

with $CO_2$ accounting for +2.16 W/m$^2$ of this increase (Forster et al., 2021), this makes $CO_2$ the most important anthropogenic greenhouse gas, accounting for 65% of the total increase in radiative forcing due to well-mixed greenhouse gases.

Soil is a critical reservoir of organic C (OC), storing ca. 3,012 Pg of OC within the surface 2 m and ca. 1,824 Pg OC in the surface 1 m (Sanderman et al., 2017). Indeed, this OC stored within soil exceeds the amount of C in the atmosphere (879 Pg) and vegetation (600 Pg C) combined, and is ca. 300-times greater than current annual emissions of C from fossil sources (9.9 Pg C, Friedlingstein et al. (2023)). Importantly, not only is the total soil organic carbon (SOC) stock large, but it is also highly dynamic – each year, ca. 61 Pg of C enters soil from vegetation whilst a similar amount is lost from soil to the atmosphere (almost entirely as $CO_2$) due to mineralization of the SOC (Lehmann and Kleber, 2015). As a result, ca. 7 % of the atmospheric C pool is cycled through soil *via* photosynthesis every year. Due to this dynamic nature of SOC, long-term disturbances to the soil can profoundly decrease global SOC stocks. In this regard, global meta-analyses have shown that long-term cropping can reduce soil OC stocks by 30-60% (Kopittke et al., 2017; Murty et al., 2002; Guo and Gifford, 2002), mainly due to lower C inputs to the soil but also to an increase in C outputs (both as $CO_2$ efflux and also outputs of biomass in the harvested product). Given the extent of global land-use change (primarily for agriculture), this loss of SOC stocks is a major global source of $CO_2$.

For any given point in time, the net global flux of $CO_2$ from soil is related to the rate of land-use change. Prior to the year 1800, the rate of land-use change was comparatively low and hence losses of SOC during this time are also estimated to be low: < 0.05 Pg C/y (50 Tg C/y) (Figure 1) (Sanderman et al., 2017). However, between 1800 and 1950, rates of land-use change increased ca. 15-fold (Klein Goldewijk et al., 2017), and as a result, losses of SOC also increased from < 0.05 Pg C/y to > 0.3 Pg C/y (equivalent to > 1.1 Pg $CO_2$/y) (Sanderman et al., 2017). Since the 1950s, rates of land-use change have decreased substantially, with the associated SOC loss also decreasing to ca. 0.1 Pg C/y between 1980 and 2000 (Sanderman et al., 2017). Thus, it is estimated that the current net global flux of $CO_2$ from soil due to SOC loss is 0.1 Pg C/y (Sanderman et al., 2017). In this regard, the current net global flux of $CO_2$ from soil is greatly surpassed by fossil $CO_2$ emissions, being 9.9 Pg C/y (Friedlingstein et al., 2023). Indeed, with total emissions of 11.1 Pg C in 2022, and assuming SOC losses are ca. 0.1 Pg C/y (equivalent to 0.37 Pg $CO_2$/y, Figure 1), the current contribution of SOC loss from land-use change accounts for only ca. 0.9% of the current annual $CO_2$ emissions (Table 1). This is because although the majority of land-use change (measured by area) has occurred over a period of a couple of hundred years and is currently decreasing, fossil $CO_2$ emissions have largely occurred during the last half-century and continue to increase rapidly.

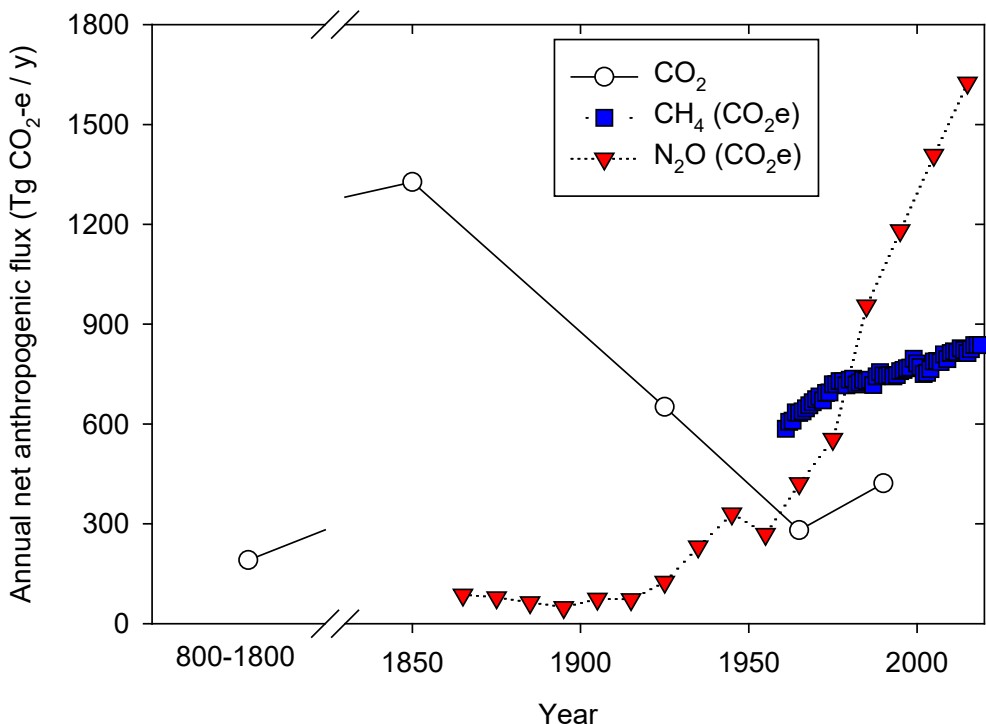

**Figure 1. Annual net anthropogenic fluxes of carbon dioxide ($CO_2$), nitrous oxide ($N_2O$), and methane ($CH_4$) from soil expressed on a $CO_2$-equivalent basis based upon their global warming potentials for a 100-y time horizon. Data points are plotted in the middle of the measurement periods (for example, the average annual emission from 1800-1900 is plotted at 1850). Data for $CO_2$ are from Sanderman et al. (2017), $N_2O$ are from Tian et al. (2019), and $CH_4$ are from Saunois et al. (2020).**

**Table 1.** Anthropogenic, soil-based emissions of greenhouse gases and their contribution to the current total increase in effective radiative forcing due to increased concentrations of well-mixed greenhouse gases.

|  | $CO_2$ | $N_2O$ | $CH_4$ |
|---|---|---|---|
| Current annual net anthropogenic flux from soil | 0.1 Pg C/y | 3.7 Tg $N_2O$-N /y | 30 Tg $CH_4$ /y |
| Current annual net anthropogenic flux from all sources | 11.1 Pg C/y | 7.3 Tg $N_2O$-N /y | 359 Tg $CH_4$ /y |
| Soil-based contribution to the current increase in effective radiative forcing (W/m²)[a] | +0.37 | +0.084 | +0.044 |
| Total current increase in effective radiative forcing (W/m²) | +2.16 | +0.21 | +0.54 |

[a] *Calculated from the proportion of total anthropogenic emissions that have been derived from soil over time whilst taking into account the atmospheric life of the gas, with this calculating the proportion of the current increase in anthropogenic radiative forcing that is due to soil.*

Next, we calculate the contribution of soil-based $CO_2$ emissions to the currently-observed increase in warming (radiative forcing) by determining the proportion of cumulative global C emissions that are from soil. In this regard, Sanderman et al. (2017) estimate that the total cumulative loss of SOC due to land-use change, together with the associated release of $CO_2$, is 116 Pg of C (425 Pg of $CO_2e$), with similar values also reported by Lal (2018). In comparison, the total, cumulative, anthropogenic $CO_2$ release from all sources from 1850-2022 is estimated to be 695 Pg of C (ca. 2,600 Pg of $CO_2e$)

(Friedlingstein et al., 2023). Thus, we estimate that the net loss of SOC due to land-use change accounts for ca. 17% of total cumulative anthropogenic $CO_2$ emissions (i.e. 116 Pg of C from the total emissions of 695 Pg of C). We note that this value is likely to be a slight over-estimate given that it accounts for total historical SOC losses but only total anthropogenic $CO_2$ emissions since 1850. Given that the total increase in radiative forcing due to increases in the well-mixed greenhouse gas concentrations is +3.32 W/m², of which $CO_2$ accounts for +2.16 W/m² (65% of the total), the release of $CO_2$ due to the loss of SOC from land-use change is estimated to account for 11% (+0.37 W/m², i.e. 17% of +2.16 W/m²) of the total increase in radiative forcing due to anthropogenic increases in well-mixed greenhouse gases (Figure 2, Table 1).

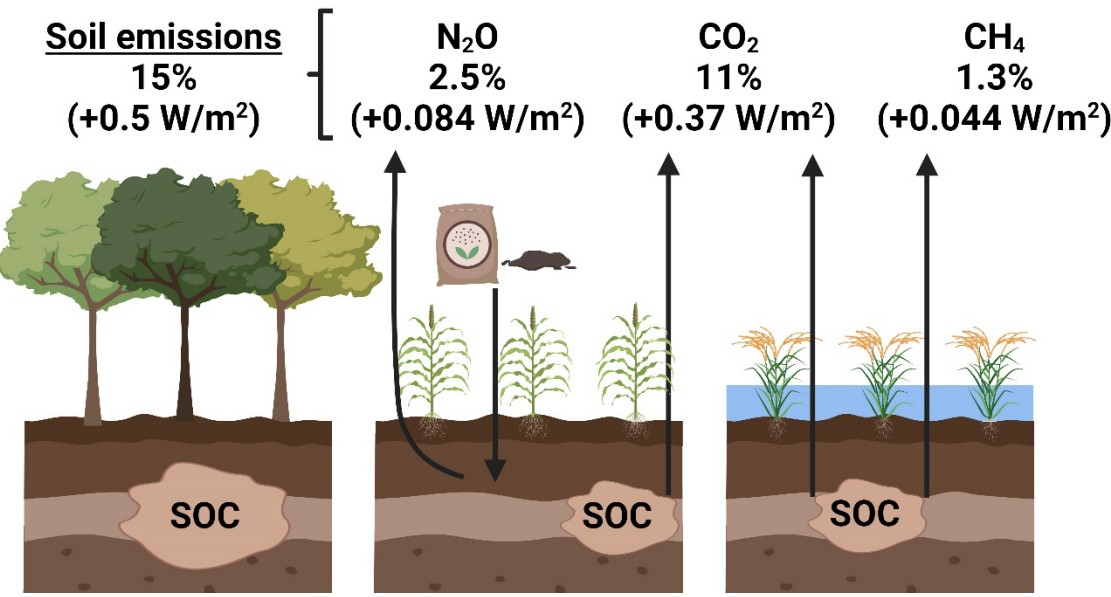

**Figure 2. Anthropogenic emissions of greenhouse gases (carbon dioxide [$CO_2$], nitrous oxide [$N_2O$], and methane [$CH_4$]) from soil and their contribution (15%, +0.5 W/m²) to the overall increase in warming (radiative forcing) due to well-mixed greenhouse gases (+3.32 W/m²). The vegetation on the left indicates native vegetation, a crop in the centre, and a rice paddy on the right.**

### 3.2 Nitrous oxide

Nitrous oxide is a greenhouse gas with a global warming potential for a 100-y time horizon that is 273-times higher than $CO_2$ (Forster et al., 2021). Atmospheric concentrations of $N_2O$ have increased from a concentration of 270 ppb in 1750 to 324 ppb in 2011, increasing a further 2.4% to 332 ppb in 2019 (Gulev et al., 2021). Of the total increase in radiative forcing due to anthropogenic release of greenhouse gases (+3.32 W/m²), +0.21 W/m² is due to $N_2O$, representing 6.3% of the total increase in radiative forcing (Forster et al., 2021).

Soil is an important source of anthropogenic $N_2O$ emissions due to increased application of reactive N (especially as inorganic N fertilizers and animal manures) and through increased use of leguminous crops. As with $CO_2$ and $CH_4$, the production of $N_2O$ in soil is a natural process, but human activities have accelerated the rate of production. Total anthropogenic emissions of $N_2O$ are estimated to be 7.3 Tg $N_2O$-N /y for 2007-2016 (Tian et al., 2020), of which, anthropogenic emissions from soil account for 3.7 Tg $N_2O$-N /y (5.8 Tg $N_2O$ /y or 1,600 Tg $CO_2$-e /y) (Figure 1) (Tian et al., 2019). Thus, soil accounts for 51% of the current anthropogenic $N_2O$ flux (Table 1). Of these anthropogenic $N_2O$ emissions from soil, croplands are of greatest concern, accounting for 82% of the soil-based increase resulting from the application of reactive N fertilizers (2.0 Tg $N_2O$-N /y), the application of manures to soil (0.6 Tg $N_2O$-N /y), and enhanced atmospheric N deposition to soil (0.9 Tg $N_2O$-N /y) (Tian et al., 2019). There has also been considerable temporal variability in the anthropogenic flux of $N_2O$ from soil, increasing

from ca. 0.2 Tg $N_2O$-N /y (87 Tg $CO_2$-e /y) in the 1860s to ca. 1 Tg $N_2O$-N /y (420 Tg $CO_2$-e /y) in the 1960s before then accelerating rapidly to the current value of 3.7 Tg $N_2O$-N /y (1,600 Tg $CO_2$-e /y, Figure 1, Table 1).

Over the period for which calculations are possible (1860s onwards) and using the second approach outlined in the Methods section, we calculate that 40% of the anthropogenic $N_2O$ currently in the atmosphere results from soil-based emissions considering the atmospheric life of $N_2O$ is 109 y. With $N_2O$ accounting for 6.3% of the total anthropogenic increase in radiative forcing, and with soil accounting for 40% of anthropogenic $N_2O$ currently in the atmosphere, we estimate that $N_2O$ emissions from soil account for 2.5% (+0.084 $W/m^2$) of the total anthropogenic increase radiative forcing due to the well-mixed greenhouse gas concentrations (+3.32 $W/m^2$) (Figure 2).

### 3.3 Methane

Methane is an important greenhouse gas with a global warming potential for a 100-y time horizon that is 27.9-times higher than $CO_2$ (Forster et al., 2021). From 2011 to 2019 alone, atmospheric concentrations of $CH_4$ increased 3.5% from 1800 to 1866 ppb, from an estimated concentration of 730 ppb in 1750 (Gulev et al., 2021). Of the total increase in radiative forcing due to anthropogenic release of well-mixed greenhouse gases (+3.32 $W/m^2$), $CH_4$ accounts for +0.54 $W/m^2$, being 16% of the total increase (Forster et al., 2021).

Soil contributes to $CH_4$ emissions primarily when waterlogged (Jiang et al., 2019). The release of $CH_4$ from soil occurs due to biogenic processes, being due to the anaerobic decomposition of organic matter. This release of $CH_4$ from waterlogged soil occurs both naturally (wetlands and swamps) and due to anthropogenic use of soil. For these anthropogenic $CH_4$ emissions, flooded rice paddies are almost entirely responsible, with rice paddies flooded to control weeds and to improve yields. Rice forms a staple food for much of the global population, with rice paddies accounting for 162 million ha of land and with rice providing an average of 18.0% of all calories consumed by humans (FAO, 2021).

Total global $CH_4$ emissions are estimated to be 576 Tg $CH_4$ /y, of which 359 Tg $CH_4$ /y is from anthropogenic sources, being 60% of the total (Saunois et al., 2020). Considering only soil-based sources, for natural emissions of $CH_4$, wetlands and swamps account for 148 Tg $CH_4$ /y, being 26% of total global $CH_4$ emissions and ca. 40% of natural sources (Saunois et al., 2020). However, for anthropogenic soil-based emissions, rice paddies are critically important, accounting for 30 Tg $CH_4$ /y (Figure 1) (Saunois et al., 2020). Thus, for current anthropogenic fluxes of 359 Tg $CH_4$ /y, soil in rice paddies account for 8% (30 Tg $CH_4$ /y, being 840 Tg $CO_2$-e /y) of the total anthropogenic emissions of $CH_4$ (Table 1) (Saunois et al., 2020).

Based upon calculations from 1980 and using the second approach outlined in the Methods section, we calculate that 8.2% of the anthropogenic $CH_4$ currently in the atmosphere results from soil-based emissions, considering the atmospheric life of $CH_4$ (11.8 y). Given that $CH_4$ accounts for 16% of the total anthropogenic increase in radiative forcing (above) and given that soil accounts for 8.2% of the anthropogenic $CH_4$ currently in the atmosphere, we estimate that $CH_4$ emissions from soil account for 1.3% (+0.044 $W/m^2$) of the total anthropogenic increase radiative forcing due to elevated greenhouse gas concentrations (+3.32 $W/m^2$) (Figure 2, Table 1).

# 4 Discussion

Soil makes a substantial contribution to net anthropogenic emissions of $CO_2$, $N_2O$, and $CH_4$, both historically and currently (Figure 1, Figure 2, Table 1). We highlight the legacy effect of historical and current human activities involving soil on climate change. Emission of $CO_2$ from soil alone accounts for 11% (+0.37 W/m$^2$) of the total increase in global warming (radiative forcing) due to well-mixed greenhouse gases, with $N_2O$ also accounting for 2.5% (+0.084 W/m$^2$), and $CH_4$ for 1.3% (+0.044 W/m$^2$). Thus, we estimate that anthropogenic use of soil accounts for 15% (+0.5 W/m$^2$) of the total increase in warming (radiative forcing) due to anthropogenic emissions of the well-mixed greenhouse gases, with $CO_2$ therefore accounting for 74% of this soil-based increase, $N_2O$ for 17%, and $CH_4$ for 8.9% (Figure 2). However, there has been substantial temporal variation – for centuries, $CO_2$ dominated net fluxes of anthropogenic greenhouse gases from soil, but comparatively recently, both $CH_4$ and $N_2O$ have overtaken $CO_2$, with $N_2O$ emissions now of particular concern (Figure 1). Urgent actions are required to protect the future climate by limiting greenhouse gas emissions from soil, such as the broad, overarching approaches indicated in Figure 3. However, to achieve these broad goals, whilst simultaneously meeting human demand for food and other products and also considering socio-economic objectives, will require the development of new and innovative approaches through to the use of incentives to encourage uptake of existing approaches by landholders (Nkonya et al., 2016).

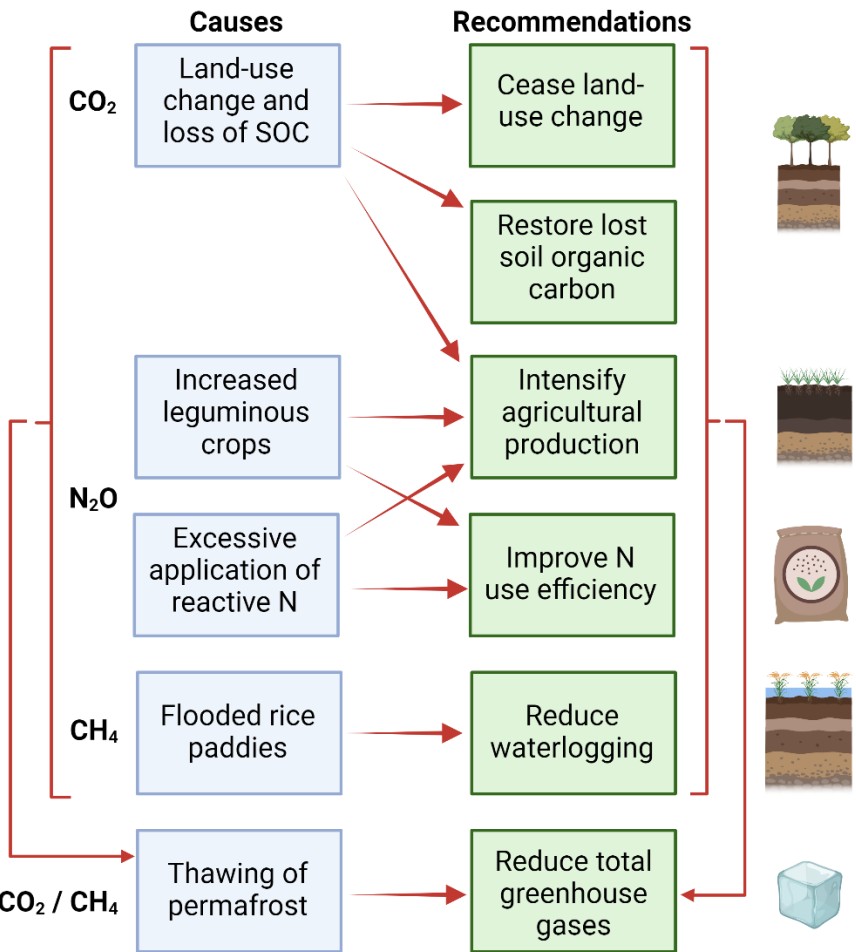

**Figure 3. Actions required to protect the future climate by limiting greenhouse gas emissions from soil. Note that 'thawing of permafrost' is related to the magnitude of greenhouse gas emissions from soil due to anthropogenic management (such as by loss of SOC from land-use change, or excessive application of reactive N), but any changes in management practices that reduce emissions of these greenhouse gases can also reduce the magnitude of this thawing and the associated release of greenhouse gases associated with the thawing of permafrost.**

## 4.1 Cease land-use change, including for bioenergy production

It is apparent that the release of $CO_2$ from soil due to loss of SOC following land-use change has had the largest adverse effect on atmospheric greenhouse gas concentrations, contributing 11% (+0.37 W/m$^2$) to the total increase in warming due to well-mixed greenhouse gases. Much of this release of $CO_2$ from soil due to land-use change is historical, having peaked between 1800 and 1900 (Figure 1), with current emissions of $CO_2$ from the loss of SOC being dominated by ongoing land-use change in 'new-world' countries such as Brazil and Argentina (Sanderman et al., 2017). [It is important to note that although the human population has increased rapidly since the 1900s, the associated increase in food production has largely not come from area expansion (land-use change), but by improving yields per unit area – the Green Revolution]. Thus, these data demonstrate that urgent emphasis must be given to ensuring that future land-use change is ceased to limit further release of $CO_2$. In particular, targeting land-use change for bioenergy production, which results in the substantial, long-term release of $CO_2$ from SOC loss. Indeed, it has been estimated that clearing of land to produce food-based biofuels creates a C debt by releasing 17-420 times more $CO_2$ than the annual reductions that the biofuels would provide by displacing fossil fuels (Fargione et al., 2008).

Cessation of land-use change is not only important for preventing further loss of SOC but also vital to protect areas where vegetation and soils are currently acting as a net sink for atmospheric $CO_2$ – the 'terrestrial sink'. In this regard, boreal and temperate forests of the northern hemisphere make the largest contribution to the terrestrial C sink (Canadell et al., 2021), with increased biomass production in these systems being largely driven by $CO_2$ fertilization and lengthening growing seasons. This is in agreement with predictions of global SOC stocks, with large areas of land, especially the boreal forests of the northern hemisphere, having a net SOC gain (Sanderman et al., 2017). Nevertheless, a potential discrepancy remains regarding the importance of SOC within the terrestrial sink. Specifically, although studies estimate that the quantity of C captured within the terrestrial C sink is currently larger than that which is lost due to land-use change (Canadell et al., 2021; Jia et al., 2019; Friedlingstein et al., 2023) leading to "increased vegetation and soil carbon" (IPCC, 2001), studies focussing primarily on SOC stocks report that net global SOC stocks are still decreasing at an average rate of ca. 0.1 Pg C/y (Sanderman et al., 2017). In this regard, it is possible that although elevated $CO_2$ may increase C within vegetation, there may not necessarily be a corresponding increase in SOC (Sulman et al., 2019), for example due to C destabilization (Bailey et al., 2019). Regardless, soil remains a net source of $CO_2$ when historical emissions are included, and hence protecting the remaining terrestrial ecosystems is vital not only to prevent the loss of SOC that results from land-use change but also because many of these systems are currently acting as a net C sink as evidenced by their increasing SOC stocks.

## 4.2 Intensify agricultural production further whilst also increasing nitrogen use efficiency

Since the 1950s and 1960s, food production has generally increased by improving yields per unit area (intensification) rather than by area expansion. Although many factors have contributed to this improved productivity, a rapid increase in the use of reactive N fertilizers as part of the Green Revolution has been critical. Indeed, through the industrial production of reactive N fertilizers, the number of humans supported per hectare of arable land has increased from 1.9 to 4.3 persons from 1908 to 2008 and with 30-50% of the increase in crop yield achieved though application of N fertilizers (Erisman et al., 2008; Stewart et al., 2005).

It is this agricultural intensification, supported by increasing rates of reactive N inputs, that has allowed rates of land-use change to slow since the 1950s with this in-turn decreasing $CO_2$ emissions from SOC loss (Figure 1). However, whilst these inputs of reactive N to cropland have enabled a decrease in $CO_2$ emissions from soil, the application of this N has concomitantly

caused a rapid increase in $N_2O$ emissions (Figure 1). Thus, whilst agricultural intensification has decreased emissions of $CO_2$ from soil, it has come at the expense of increasing $N_2O$ emissions – a potent greenhouse gas (Figure 1).

To limit future land-use change whilst simultaneously increasing global food production will require even further intensification of agriculture (Kopittke et al., 2019). Thus, given the already rapidly increasing emissions of $N_2O$ from soil, it is imperative that strategies should be targeted through sustainable intensification (Pretty and Bharucha, 2014; Pretty et al., 2018), and that they should be developed and implemented to increase N use efficiency and decrease $N_2O$ emissions. This can be achieved by more closely aligning N supply to plant demand, such as through the repeated (multiple, strategic) applications of N fertilizer during the growing season, through the development of improved genotypes with higher N use efficiency, and through the use of slow-release fertilizers (Snyder et al., 2014). Increasing N use efficiency also has the additional advantages of decreasing soil acidification and environmental eutrophication whilst also the additional agronomic benefit of increasing farmers profitability. Of course, care must also be taken to ensure that this intensification of production (in order to limit future land-use change) does not increase overall C losses, with changes in C gains and losses depending upon a range of factors such as agronomic practices, environmental conditions, and soil properties.

### 4.3 Decreases in the methane flux from soil can rapidly decrease radiative forcing

Although soil-based emissions of $CH_4$ contribute a more modest 1.3% (+0.044 W/m2) to radiative forcing (Fig. 1 and Fig. 2), decreasing the rate of $CH_4$ emission from soil would yield a comparatively rapid decrease in radiative forcing given the atmospheric life of $CH_4$ is only 11.8 y. In this regard, decreasing the emission of 30 Tg $CH_4$ per year from the soil of the 162 million ha of rice paddies globally can potentially be achieved by reducing the period of time that the soil is waterlogged, with midseason drainage and intermittent irrigation known to reduce $CH_4$ emissions by up to 90% (Souza et al., 2021; Islam et al., 2018). A range of other approaches have also been reported to be effective at reducing $CH_4$ emissions from rice paddies (Nikolaisen et al., 2023; Hussain et al., 2015). For example, organic amendments such as straw should only be applied during aerobic conditions (off-season) rather than during on-season application (Nikolaisen et al., 2023). Such approaches are critical in maintaining yield whilst also decreasing $CH_4$ emissions from soil (Smith et al., 2021). Furthermore, in many areas, strategies that will reduce $CH_4$ emission from flooded rice culture will also deliver the benefit of increased water use efficiency.

### 4.4 Avoiding future thawing of permafrost

There is increasing concern regarding the release of $CO_2$ due to the accelerating thawing of permafrost C in the Arctic and sub-Arctic. This permafrost contains ca. 1,035 Pg of C to 3 m depth (Schuur et al., 2015), with a warming climate causing increased thawing of the permafrost and the associated release of $CO_2$ (and $CH_4$). Indeed, it is estimated that ca. 92 Pg of C (ranging from 37-174 Pg of C) is susceptible to release as greenhouse gases in the present century (IPCC, 2019). Of these future C emissions from permafrost, it is expected that $CO_2$ emissions will account for ca. 98% of the C, with $CH_4$ expected to account of 2.3% of future emissions (Schuur et al., 2013). Whilst this proportion projected to be released as $CH_4$ is comparatively small (2.3%), given that $CH_4$ has a global warming potential for a 100-y time horizon that is 27.9-times higher than $CO_2$, this equates to an increased warming potential of this permafrost C of 35-48% when accounting for the increased potency of the $CH_4$ (Schuur et al., 2015). Thus, although the release of $CO_2$ due to loss of SOC from land-use change has now slowed, the predicted release of $CO_2$ from the thawing of permafrost (ca. 90 Pg of C) during the present century alone is of similar magnitude to the cumulative emissions of $CO_2$ over the last > 1000 years [116 Pg of C, Sanderman et al. (2017)] due to progressive land-use change, with this being a positive carbon-climate feedback. Thus, limiting the extent of future climate change is essential in preventing the profound release of $CO_2$ and $CH_4$ from permafrost.

### 4.5 Restore a portion of the soil organic carbon that has been lost historically

Of the 116 Pg of C that has been lost from soils historically (Sanderman et al., 2017), a portion can be restored by implementing best soil management practices on croplands and grazing lands. Smith et al. (2020) estimate a technical potential for soil carbon sequestration of up to 2.2 Pg C $y^{-1}$ globally, with an economic potential of 0.4-0.7 Pg C $y^{-1}$ (Smith et al., 2008). Assuming that a new equilibrium is reached in 20 y as per IPCC Good Practice Guidance, this gives a cumulative maximum technical potential of 44 Pg C, and an economic potential of 8-14 Pg C for restored SOC.

### 4.6 A broad portfolio of responses is required, including actions not related directly to soil

Solving the challenges associated with anthropogenic greenhouse gas emissions from soil will be highly complex, with the first step being to describe and quantify the underlying problems (as we have done in the present study) so that mitigation approaches can be targeted and specific. To this point, we have discussed soil-based approaches that address the individual challenges (such as land-use change, improving N use efficiency, and so forth). However, it is clear that a multitude of responses will be required and not single approaches in isolation.

Additionally, we cannot solely focus on the biophysical issues as discussed above, but consideration also must to be given to socio-economic and institutional aspects which are also critically important (Nkonya et al., 2016). For example, although a range of management practices and technologies already exist to address some of these problems, many are not implemented fully, often due to economic drivers and the desire to maximize profits and minimize costs (Boardman et al., 2003). Consider, for example, that N fertilizer efficiency can often be improved by use of slow-release sources or by application of a larger number of smaller doses of N fertilizer to more closely match plant demand (c.f. a fewer number of larger doses), but the increased costs associated with such practices means that they are often not implemented to the extent possible. In a similar manner, rural poverty is also an important driving factor, with many of the rural poor farming highly marginal lands where the necessity to provide food for their families overrides concerns for soil degradation and environmental harm.

There are also a range of other non-soil-based approaches that are not considered in detail here that could also make substantial contributions. For example, it is estimated that one-third of all food produced for human consumption is lost and wasted each year (Ishangulyyev et al., 2019), and hence reducing wastage could substantially reduce the area of land required for food production.

### 4.7 Limitations and uncertainties of study

In the present study, we have gathered the best available global estimates of greenhouse gas emissions from soil from across multiple studies (Sanderman et al., 2017; Tian et al., 2019; Saunois et al., 2020), with each of these studies having various assumptions and uncertainties that are carried forward.

First, we acknowledge that the data presented within these previous studies have uncertainty and that these uncertainties influence the calculations presented within the present study. However, we are unable to include uncertainties in our calculations given that some of the previous studies themselves did not report them. For example, Sanderman et al. (2017) report that SOC loss is 116 Pg C, but these authors do not report a measure of uncertainty or error with this value. Thus, when calculating the contribution of soil-derived $CO_2$, $N_2O$, and $CH_4$ to anthropogenic greenhouse gases, we are unable to provide error estimates for their total contribution. Regardless, it is important to note that the value of 116 Pg C reported by Sanderman

et al. (2017) for agricultural land use change, for example, is similar to previously-reported values, such as 115-154 Pg C reported by Lal (2018), 85 Pg C reported by Padarian et al. (2022) for cropping alone (i.e. excluding pasture and grasslands), and 80-100 Pg C by Lal (1999) for conversion of natural to managed systems. Whilst our inability to include measures of uncertainty is a substantial limitation of our study, this reflects the observation that additional work is urgently required to allow for more rigorous assessments given the importance of soil-based emissions of greenhouse gases.

As a second limitation, we acknowledge the different time periods examined by the studies we have utilized for $CO_2$ (> 1000 y) $N_2O$ (> 100 y) and $CH_4$ (ca. 40 y). This limitation arises from differences between the three studies upon which we have based the present assessment. However, we do not consider that this is an important limitation. For $N_2O$, we include data from the 1860s onwards, with anthropogenic emissions prior to this time being negligible (Tian et al., 2019), whilst for $CH_4$, although the data are available only for the last ca. 40 y, given that the atmospheric life of $CH_4$ is 11.8 y, emissions of $CH_4$ prior to this time would not make substantial contributions to current $CH_4$ concentrations in the atmosphere. Thus, despite the timeframes being markedly different between these previous studies, we contend that this does not markedly influence the outcome.

Third, it is important to note differences within the scope of the three studies upon which the present one is based. For example, the study of Sanderman et al. (2017) only examined agriculture and hence did not consider other forms of land use change such as urban development on the loss of SOC. Regardless, given that urban areas consisting of ca. 0.7% of global land surface (Zhao et al., 2022) whilst cropping accounts for 12% of the ice-free land and permanent grassland and pasture account for 25% (FAO, 2021), the magnitude of this error is likely small.

Fourth, we make the assumption here that all SOC that is lost from soil has been emitted to the atmosphere as $CO_2$ through mineralization. We expect that the error from this assumption is only small given that during the early stages (up to ca. 20 y) following land-use conversion, when the loss of SOC is the greatest, 80% of the SOC is lost due to mineralization whilst 20% to erosion (Lal, 2001). Furthermore, even for SOC which is eroded rather than directly mineralized, the majority of this eroded SOC is simply redistributed to other soil within the landscape – of the 5.7 Pg C eroded by water annually, 3.9 Pg C is simply redistributed over the landscape, 0.57 Pg C is buried in lakes and reservoirs, and 1.14 Pg C is mineralized (Lal, 1995). Thus, whilst not all SOC that is lost from soil is mineralized to the atmosphere as $CO_2$, the error is likely to be small.

Despite the clear limitations noted above, we collate these data as a starting point to highlight and discuss the critical importance of anthropogenic management of soil on greenhouse gas emissions. In this regard, it is imperative that future studies refine these estimates with more comprehensive data. Regardless, despite these uncertainties, the relative importance (contribution) from each greenhouse gas over different timescales is unlikely to differ greatly from those presented here, even once better data become available.

In the present study we have focused on quantifying the overall contribution of soil to global anthropogenic greenhouse gas emissions and global climate change, comparing $CO_2$, $N_2O$, and $CH_4$. By comparing the historical and current global emissions between these three gases, it has been possible to identify important trends over time, with this being critical for focussing future efforts. It is important to note, however, that there is also spatial variability in these emissions across the planet. For example, it is known that some soils (particularly boreal and temperate forests of the northern hemisphere) are making an increased contribution to the terrestrial (vegetation plus soil) C sink (Canadell et al., 2021; Sanderman et al., 2017). Thus, developing detailed plans to mitigate C emissions (or other greenhouse gases) must first consider such variability.

Finally, it must be noted that in the present study we have focused on the release of greenhouse gases due to anthropogenic management of soil and we have not considered the role of soil in removing $CO_2$ from the atmosphere and by SOC. Indeed, there are an increasing number of studies examining role of soil as a 'negative emission technology' (NET) for the capture of $CO_2$ from the atmosphere (Smith, 2012; Paustian et al., 2016; Minasny et al., 2017; Lal et al., 2021; van Vuuren et al., 2018). However, for soil to be effective as a NET, we must first reduce the substantial emissions of greenhouse gases from soil.

## 5 Conclusions

Our increasing focus on soil to provide biomass (especially food) for human use through intensive agriculture has caused soil to release profound amounts of greenhouse gases, with this threatening planetary survivability. We show that anthropogenic emissions of greenhouse gases from soil account for 15% of the entire global increase in warming (radiative forcing) caused

by well-mixed greenhouse gases. Although $CO_2$ is the most important greenhouse gas emitted from soil (74% of the total warming), much of this $CO_2$ has been emitted historically with current rates of release considerably lower. Thus, for $CO_2$, efforts should be directed towards limiting the rate of release increasing again by preventing ongoing land-use change, restoring a portion of the historically lost soil organic carbon through soil best management practices, whilst also preventing global warming that will result in release from permafrost. However, to prevent further land-use change will require ongoing

intensification of production through increased N fertilizer application, with strategies required to markedly improve N fertilizer efficiency and limit the already rapidly accelerating emissions of $N_2O$ – a potent greenhouse gas. We also need to decrease $CH_4$ emissions from rice paddies, which although may only have a comparatively modest impact, has the advantage that reduced emissions result in a benefit in the short term. Finally, although the present study highlights how human-use of soil is resulting in substantial releases of greenhouse gases and contributing to global warming, it is also important to note that

soil also acts as a sink for greenhouse gases (whether emitted from soil or from other anthropogenic sources), with this also being a critical role of soil. Recognizing the central importance of soil in contributing to climate change is essential if we are to maintain planetary hospitability.

## 6 Author contributions

Conceptualization, P.M.K., R.C.D., B.A.McK., P.W.; Methodology, P.M.K., R.C.D., P.S., N.W.M., Formal analysis, P.M.K., R.C.D., B.A.McK., P.W., Z.W., F.J.T.vdB.; Writing, P.M.K., R.C.D., B.A.McK, P.S., P.W., Z.W., F.J.T.vdB., N.W.M.

## 7 Data availability

The data and analyses that support these findings will be made available in response to a reasonable request.

## 8 Supplement

The supplement related to this article is available online at: Xxxxxx

## 9 Competing interests

The authors declare that they have no conflict of interest.

## 10 Disclaimer

Publisher's note: Copernicus Publications remains neutral with regard to jurisdictional claims made in the text, published maps, institutional affiliations, or any other geographical representation in this paper. While Copernicus Publications makes every effort to include appropriate place names, the final responsibility lies with the authors.

## 11 Acknowledgements

The authors acknowledge the assistance of Yriah Rusden.

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
