# Peer review of "Soil is a major contributor to global greenhouse gas emissions and climate change"

_EGUsphere, 2024_

## Author Response (AR1)

**Reviewer 1**

*General Comments*

*The manuscript titled 'Soil is a major contributor to global greenhouse gas emissions and climate change' takes a broad view of the role of anthropogenic land use change, especially with regard to agricultural lands used to grow food, in global soil GHG emissions and soil carbon loss. The authors base their analysis on the previous studies of Sanderman et al. (2017) for CO2, Tian et al. (2019 and 2020) for N2O, and data from FAO (2021) for CH4 emissions. This is a valuable global analysis and synthesis that finds that net anthropogenic emissions from soil alone account for 15% of the entire global increase in radiative forcing caused by well-mixed greenhouse gases, with CO2 being the most important gas emitted from soil (74% of total soil-derived warming) followed by N2O (17%) and CH4 (9%). The authors suggest that there's an urgent need to prevent land use change to the best of our ability and to specifically take action to prevent further thawing of permafrost, to decrease rice paddy methane emissions, and to increase N fertilizer efficiency.*

We thank this reviewer for providing us with feedback and for assisting us in improving the manuscript. We also thank the reviewer for noting that this is a "valuable global analysis and synthesis".

*However, there are a number of concerns regarding mismatch between study findings and recommendations. The authors have not fully considered other important soil C loss mitigation strategies or a more targeted approach for the most vulnerable regions given their capacity for soil C gains or GHG emissions. Current frameworks for soil carbon management are not fully explored.*

We agree that we have not examined this in detail, but nor is this the focus of our study. Rather, before it is possible to develop appropriate and targeted C management approaches, it is first imperative that the contribution of soil to emissions of the various greenhouse gases be fully quantified. Without this information, it is not possible to develop appropriate carbon management strategies, or indeed, ensure that our focus is on the most pressing needs. For example, as illustrated in the present study, greenhouse gas emissions are currently much more substantial for nitrous oxide than for carbon dioxide, and in terms of reducing ongoing emissions, the focus should actually be on nitrous oxide rather than carbon – this is the value of our current approach in subsequently enabling targeted management approaches. This is described in the Introduction, for example, where we state: "in order to improve management practices and to inform better decision-making processes, it is imperative that we quantify the precise sources of greenhouse gases and understand the factors causing their emissions".

*The authors discuss the reason for excluding uncertainty estimates, but I am not convinced that this wasn't possible given that the main data they reference does include confidence bounds.*

The data originally published by Sanderman et al (2017) erroneously lists the value as being 133 Pg C, but this was modified in the "corrected" version of their manuscript to be 116 Pg C (DOI: 10.1073/pnas.1800925115). In this corrected version, we are unable to see any confidence bounds and hence we are unable to include it in our present study. Furthermore, even in the original study (DOI: 10.1073/pnas.1706103114), we are unable to see any confidence intervals in the

original text for the value of 133 Pg C, nor any confidence intervals for the historical values shown Figure 2 (for example, also refer to Table S3 and Figure S10 of Sandermann et al 2017). If we are mistaken, we would appreciate the Reviewer assisting us with this information.

*They suggest a multifaceted approach to land management to grow the food needed to support the global population, but have failed to enumerate any innovative or targeted approaches and instead suggest a few broad, sweeping needs that were not investigated by the present study and that have not been critically assessed for their feasibility of implementation.*

Please see detailed response above. Specifically, it was the aim of our study to first quantify the emissions of greenhouse gases from soil – this information is essential if we are to subsequently develop innovative or targeted approaches.

*Specific Comments*

*Abstract should include a statement about the data sources used for this analysis.*

We have modified the Abstract to note that external data sources were used, but we note that it is against the rules of the journal to include references in the Abstract.

*Regarding study methods: See Crow & Sierra (2022): The climate benefit of sequestration in soils for warming mitigation. Biogeochemistry, 161(1), 71-84. for an alternative, potentially more robust, computational framework that assesses the contribution of simultaneous emissions and uptake on radiative forcing as the 'climate benefit of sequestration'.*

We thank the Reviewer for highlighting this important study, which we have now cited in our manuscript (Line 71). However, the focus of Crow and Sierra (2022) is different to ours and we are not sure how their approach would benefit in our study. This is because we are not seeking to model changes over time. Rather, in our study, we take previously-computed values for greenhouse gas concentrations emitted from soil (with these previous models already examining changes over time), and based upon these values, we determine the contribution to radiative forcing.

*Sanderman (2017) reports uncertainty estimates, so why not use those in the current modeling effort? There is, potentially, substantial uncertainty in these estimates given the mismatch between model estimates (Sanderman 2017) and measured SOC given known spatial and temporal complexity of SOC and GHG pools and processes.*

See comment above.

*L165: C outputs as CO2 efflux only or outputs to biomass vegetation too?*

We have clarified the text to avoid this confusion (Line 171). Specifically, the increase in outputs is both as the increased mineralization to $CO_2$ (heterotrophic respiration) but also in the export of products for human consumption.

*L178-180: Clarification needed regarding land-use change decreasing in terms of acreage of land affected globally or intensity of the change? Thinking about rainforest deforestation and extreme ecosystem effects relative to other types of land use changes, for example.*

We have clarified the text as suggested (Line 184).

*Fig. 2: Add to the figure caption a brief note of the ecosystem/vegetation and features that each panel represents or label them below each panel. Figure takes up a lot of space and it's not strictly necessary to show the entire soil profile, since the C and other GHGs are mainly being lost from the top of the profile.*

Figure 2 was modified as suggested.

*L280 suggests that a multifaceted approach is needed that includes landowner incentives and points to Fig 3, but the figure does not include any description of how these changes might be accomplished. They are very broad recommendations and do not include any novel or innovative approaches.*

The focus of the present study was not to develop novel management approaches. Rather, our focus was to "quantify the substantial quantities of anthropogenic greenhouse gas emissions from soil and their contribution to climate change", with this information being critical in subsequently developing targeted management approaches. Indeed, if we do not first understand the nature of greenhouse gas emissions from soil, we are unable to develop and implement appropriate management approaches.

*L 286: Regarding the recommendation to cease land-use change, including for bioenergy production. Is there enough evidence from the current study to support this recommendation? I only see one older reference to support this statement (Fargione et al., 2008) and this is not something that the present study measured, so it is not appropriate to conjecture about. Blanket statements such as this should be made with caution, since there is a body of evidence which supports targeted use of certain marginal lands for biofuel production (but of course not something like clearing rainforests to grow biofuel feedstocks). I recommend a more measured argument based on evidence from the present study.*

We contend that there is enough evidence to support this – widespread land-use has caused profound release of greenhouse gases historically (Figure 1), and so we need to avoid rates of land-use change increasing again which would be associated with a concomitant release of greenhouse gases.

*Fig 3. For the recommendation to reduce waterlogging, how feasible is this in rice paddy production systems? Is it more feasible in some than others? Explore this idea more fully using support from other studies if needed. I don't understand the arrow connecting thawing of permafrost to the set of recommendations. Wouldn't it instead make sense to connect 'reduce total GHGs' to the set of other recommendations? Otherwise it would make sense to connect more of the boxes to each other in perhaps three tiers instead of 2, since thawing of permafrost is caused by the increased GHGs that result from the other land management actions listed as causes.*

We have modified the text (Section 4.3) to include additional information and additional references to highlight that it is possible to decrease methane emissions. In addition, we have modified the arrows in Figure 3 and thank the reviewer for noting this problem.

*L 310-311: Consider also referencing the publication: Bailey, V. L., Pries, C. H., & Lajtha, K. (2019). What do we know about soil carbon destabilization?. Environmental Research Letters, 14(8), 083004.*

We have now referenced this publication as suggested (Line 320).

*L 316: This argument about intensifying agricultural production has been made frequently before, but what are the bounds we can reasonably do this within? How does increasing the intensity of land use, rotation frequency, planting density, etc. influence the balance between C losses and gains? How much does this depend on the system, ecology, environment, and previous land use conditions? In my mind these two statements should be considered separately.*

We have modified the manuscript according to the comments from the Reviewer (Line 347-349) to note that this is a balance and that it depends upon the system, environment, and land-use conditions.

*What about studies that have considered the diversion of food waste and argue that we already produce enough food globally; we just waste some large amounts?*

The Reviewer makes an important point which we have now incorporated into the revised manuscript (Line 402-405).

*What about arguments that a portfolio of approaches are necessary to address the magnitude of our current problem? Locally produced food, reducing emissions from importing food (food miles), addressing food waste, incentivizing agricultural and land management practices that build soil carbon and lessen emissions, simultaneously increasing N-use efficiency through precision agricultural technologies, reducing barriers to access to healthy food for marginalized communities, etc.*

We thank the Reviewer for noting this, which we have now incorporated into the revised manuscript as an entire section (Section 4.6).

*L418-422: I appreciate this perspective, but given the spatiotemporal variability of soil C losses and gains, it is important to consider the inputs and outputs when possible. What about the Sanderman (2017) perspective that there are identifiable regions that should be targets for soil C restoration efforts? Managing soil C effectively requires taking its differences into account - in terms of past and current land use, climatic differences, land management goals and potential land use, state of degradation, current ecosystem function and services, benefits, vulnerability to C loss, etc.*

We have now modified this statement to avoid confusion (Line 455-459). Specifically, the approach we have taken does indeed account for existing gains in C in some regions (for example, Boreal forests) as this is part of approach used by Sandermann et al in their study. Rather, we were attempting to convey that our focus

was on quantifying the historical and current net emissions of greenhouse gases rather than developing novel approaches that could be used in the future for sequestering carbon in soils.

*Technical Corrections*

*L170. Remove error message: (Figure 1Error! Reference source not found.)*

We have now removed this error message.

*L 235. "considering the atmospheric life of N2O is 109 y"*

We have fixed this grammatical error.

**Reviewer 2**

*Kopittke et al. quantified the contribution of soil to global anthropogenic greenhouse gas emissions. The results showed net anthropogenic emissions from soil alone account for 15% of the entire global increase in climate warming (radiative forcing), with carbon dioxide being the most important gas emitted from soil (74%), followed by nitrous oxide (17%) and methane (9%). Moreover, limiting the release of carbon dioxide that results from loss of soil organic carbon, to develop strategies to increase nitrogen fertilizer efficiency, to decrease methane emissions from rice paddies, and to ensure that the widespread thawing of permafrost is avoided, are effective in reducing greenhouse gas emissions from soil. The results from this study are interesting and important in the era of global climate change, however, the manuscript was not well-prepared and should be revised carefully.*

We appreciate the time that the reviewer has taken to read our manuscript and provide us with this valuable feedback.

*The Introduction should clearly show the novelty and importance of the study.*

We thank the Reviewer for this suggestion. We have now revised the manuscript to more clearly articulate the novelty of this work, including a clearer delineation of 'soil' and 'land' given that previous studies have focused on land but here we distinguish here between 'soil' and 'land' as is required for understanding, valuing, and managing soil as a discrete component of land (Line 59-75).

*The Methods section is not clear, please describe how to obtain and analyze the data in detail to support the robust results.*

We apologize if something is not clear. If the Reviewer is able to provide specific information, we will make changes accordingly. However, it is not apparent to us what information in the Methods section is not clear.

*The "Results" are your results, not the others (e.g. lines 149-150). There are many references cited in this part.*

We have carefully examined the Results section but we are unclear as to what changes can be made. The references that are cited have not been cited to provide a comparison with our data. Rather, these references are cited as they provide the actual data (source data) that we are analyzing.

*The "Discussion" should be the discussion of results from this study by comparing the published literature.*

We agree with the reviewer although we are unsure exactly what is being requested. If the Reviewer is able to provide specific details, we will modify the text as appropriate.

*Spelling and grammatical issues should be carefully checked (e.g. Line 170).*

We have checked the text carefully to fix any spelling and grammatical errors, including the one identified by the Reviewer.

*While I think this study is interesting, however, the manuscript should be well-organized and presented, and revisions are needed to make it more readable, logical, and credible.*

We agree with the reviewer and we have checked the manuscript carefully. However, we are unsure why the reviewer states that revisions are required to make it more "credible" and without specific details we are unable to make revisions in this regard.